# Effects of Probiotic NVP-1704 on Mental Health and Sleep in Healthy Adults: An 8-Week Randomized, Double-Blind, Placebo-Controlled Trial

**DOI:** 10.3390/nu13082660

**Published:** 2021-07-30

**Authors:** Hyuk Joo Lee, Jung Kyung Hong, Jeon-Kyung Kim, Dong-Hyun Kim, Seok Won Jang, Seung-Won Han, In-Young Yoon

**Affiliations:** 1Department of Psychiatry, Uijeongbu Eulji Medical Center, Eulji University, Gyeonggi 11759, Korea; hjlee84.hl@gmail.com; 2Department of Neuropsychiatry, School of Medicine, Eulji University, Daejeon 34824, Korea; 3Department of Psychiatry, Bundang Hospital, Seoul National University, Seongnam 13620, Korea; koalalala@nate.com; 4Neurobiota Research Center, College of Pharmacy, Kyung Hee University, Seoul 02447, Korea; kim_jk0225@naver.com (J.-K.K.); dhkim@khu.ac.kr (D.-H.K.); 5PB Department, Navipharm Inc., Suwon 16209, Korea; swjang@navipharm.co.kr (S.W.J.); swhan@navipharm.co.kr (S.-W.H.); 6Department of Psychiatry, School of Medicine, Seoul National University, Seoul 03080, Korea

**Keywords:** probiotic, gut microbiome, depression, anxiety, sleep

## Abstract

The human gut microbiome is closely linked to mental health and sleep. We aimed to verify the efficacy and safety of probiotic NVP-1704, a mixture of *Lactobacillus reuteri* NK33 and *Bifidobacterium adolescentis* NK98, in improving stress, depression, anxiety, and sleep disturbances, along with the measurement of some blood biomarkers. A total of 156 healthy adults with subclinical symptoms of depression, anxiety, and insomnia were retrospectively registered and randomly assigned to receive either NVP-1704 (*n* = 78) or a placebo (*n* = 78) for eight weeks. Participants completed the Stress Response Inventory, Beck’s Depression and Anxiety Inventory, Pittsburg Sleep Quality Index, and Insomnia Severity Index at baseline, at four and eight weeks of treatment. Pre- and post-treatment blood tests for biomarkers were conducted. After intervention, gut microbiota composition was quantified by pyrosequencing the bacterial 16S rRNA gene. The NVP-1704 group had a more significant reduction in depressive symptoms at four and eight weeks of treatment, and anxiety symptoms at four weeks compared to the placebo group. Those receiving NVP-1704 also experienced an improvement in sleep quality. NVP-1704 treatment led to a decrease in serum interleukin-6 levels. Furthermore, NVP-1704 increased *Bifidobacteriaceae* and *Lactobacillacea*, whereas it decreased *Enterobacteriaceae* in the gut microbiota composition. Our findings suggest that probiotic NVP-1704 could be beneficial for mental health and sleep.

## 1. Introduction

The gut microbiota consists of a community of microorganisms including bacteria, viruses, protozoa, and fungi in the gastrointestinal tract, and is defined as the totality of these microbes and their genomic components [1]. A substantial body of work indicates that the human gut microbiome is implicated in mental health via bidirectional interactions within the brain–gut–microbiome (BGM) axis, which involves the immune, neural, endocrine, and metabolic pathways between the gut and the brain [2,3,4]. The perturbation of this axis leads to altered stress response and behaviors, and has been proposed to be involved in several psychiatric disorders, including depression [5], anxiety [6], and insomnia [7].

The composition and activity of the gut microbiota can be modulated by dietary intake [8]. Diet modification rapidly alters the gut microbial community [9], and a complex and varied diet is associated with a more diversified gut microbiome [10]. From a clinical perspective, diet-induced improvements in the diversity and function of the gut microbiome have the potential to improve mental health by affecting the BGM axis. Thus, the oral intake of probiotics, a preparation of live beneficial microorganisms [11], may have therapeutic effects on psychiatric symptoms by promoting a healthy and balanced gut microbiome.

Probiotics contain a variety of microbes and may exert varying health effects depending on their microbial composition. In recent years, accumulating literature suggests that probiotics composed of specific strains of genera *Lactobacillus* and *Bifidobacterium* may have the potential to prevent and treat various psychiatric conditions such as depression and anxiety [12]. However, to the best of our knowledge, only a few randomized controlled clinical trials have investigated the effects of probiotics containing *Lactobacillus* and *Bifidobacterium* strains on mental health in healthy individuals, and these studies have produced conflicting results [13,14,15]. Furthermore, these studies had small sample sizes and did not conduct gut microbiome analyses to confirm the restoration of homeostasis of the gut microbiome, one of the most reliable markers of probiotic effects [16].

The probiotic NVP-1704 is a mixture of *Lactobacillus reuteri* NK33 and *Bifidobacterium adolescentis* NK98 isolated from the healthy human gut microbiota [17], and several preclinical trials have showed that NVP-1704 can alleviate anxiety and depression in mice by modulating gut immune responses and gut microbiota composition [17,18]. Translational research to confirm these psychotropic effects of NVP-1704 may be the first step to providing a novel therapeutic approach to alleviate psychiatric symptoms.

The present study aimed to examine the efficacy and safety of NVP-1704 administration for the management of stress-related symptoms, such as depression, anxiety, and insomnia, in healthy adults. To overcome the methodological shortcomings of previous studies, we conducted a randomized, double-blind, placebo-controlled parallel study with a relatively large sample size, and included a gut microbiome analysis to confirm the effects of our intervention.

## 2. Materials and Methods

### 2.1. Study Population

The present study was conducted at the sleep clinic of Seoul National University Bundang Hospital. Healthy adults aged 19 to 65 years with psychological stress and subclinical symptoms of depression or anxiety were recruited by advertising in the hospital and the local community, from October 2018 to August 2019. Enrolled participants underwent screening tests at the first visit (within two weeks prior to initiation of the intervention), and we included those who met both of the following criteria: (1) Stress Response Inventory (SRI) score ≥ 50 and ≤ 100; and (2) either Beck Depression Inventory-II (BDI-II) score ≥ 20 and ≤ 45 or Beck Anxiety Inventory (BAI) score ≥ 16 and ≤ 45. The Korean version of SRI [19], BDI-II [20], and BAI [21] were developed and validated in the Korean population. The exclusion criteria were as follows: (1) psychiatric disorders including mood disorders, anxiety disorders, and psychotic disorders; (2) withdrawal syndrome from alcohol or smoking; (3) history of treatment for stress, depression, and anxiety within four weeks prior to study; (4) any severe stressful life events within four weeks prior to study; (5) use of oral steroids, sleeping pills, anorectic agents, beta-blockers, antibiotics, drugs related to colon diseases, any type of pre- or probiotics including yogurt and other functional foods for stress or insomnia within two weeks; and (6) medical illness such as uncontrolled hypertension, uncontrolled diabetes mellitus, thyroid diseases, and impaired renal or hepatic function. All participants agreed to participate in the study and signed informed consent before the initiation of the study. This study was approved in August 2018 by the Institutional Review Board of Seoul National University Bundang Hospital (B-1807/483–005) and was registered retrospectively in the Clinical Research Information Service, Korea (registration number KCT0004801).

### 2.2. Intervention

In the present study, the probiotic NVP-1704 was chosen as the active agent. NVP-1704 is composed of *Lactobacillus reuteri* NK33 and *Bifidobacterium adolescentis* NK98 freeze-dried with maltodextrin. Each 500 mg capsule of NVP-1704 contained 2.5 × 10^9^ colony-forming units of microorganisms (2.0 × 10^9^ CFU for *Lactobacillus reuteri* NK33 and 0.5 × 10^9^ CFU for *Bifidobacterium adolescentis* NK98). The placebo capsule, made of maltodextrin, was created to look identical to the NVP-1704 capsule. All study products were prepared at a Good Manufacturing Practice-certified manufacturing facility and stored at 2–8 °C. The participants were instructed to keep the products refrigerated.

At the first visit, demographic, medical, and anthropometric data were gathered, and electrocardiogram, blood cell count, biochemistry profile, and urinalysis were conducted for each participant. At the second visit, two weeks after the first visit, eligible participants were randomly assigned in a 1:1 ratio to either the experimental group receiving the probiotic NVP-1704, or the control group receiving a placebo. The random assignment was performed via the block randomization method, using SAS version 9.4, and both participants and investigators were blinded to the treatment conditions. Participants received either the probiotic NVP-1704 or a placebo depending on their randomized group assignment, and intervention began immediately. They were instructed to take two capsules with water once a day, daily, for eight weeks. The third and final visits took place at four and eight weeks after initiation of intervention, respectively. Treatment compliance was assessed twice over the study period by counting the remaining capsules at the last two visits. Figure 1 describes the timeline of the study protocol.

### 2.3. Questionnaires

All participants were asked to complete self-report questionnaires regarding symptoms of stress, depression, and anxiety at the first, third, and final visits. The SRI was adopted to assess emotional, somatic, cognitive, and behavioral stress responses. The SRI consists of 39 items with total scores ranging from 0 to 156, with higher scores indicating more severe stress symptoms [19]. Symptoms of depression and anxiety were evaluated using the BDI-II [22] and BAI [23], respectively. As an additional analysis, overall depression and anxiety symptoms were evaluated by the sum of the BDI-II and BAI scores, because depression and anxiety tend to co-occur and often overlap in phenomenology and pathophysiology [24].

Subjective sleep quality and symptoms of insomnia were assessed using the Pittsburgh Sleep Quality Index (PSQI) and Insomnia Severity Index (ISI), respectively. The Korean version of PSQI [25] and ISI [26] were developed and validated in the Korean population. The PSQI consists of seven subscales that assess various aspects of sleep quality; we performed sub-analyses for each subscale score of the PSQI to identify the specific effects of our intervention. We also compared the rate of clinically meaningful improvements in sleep quality and insomnia symptoms between the two groups for additional analysis. Clinically meaningful improvements were defined as a score reduction of more than or equal to the minimal clinically important difference (MCID), which refers to a threshold value for a clinically meaningful and worthwhile change [27]. Based on several previous studies, we determined the MCID for the PSQI [28,29] and ISI scales [30] as 3 and 6 points, respectively.

### 2.4. Blood Biomarkers

The pre- and post-intervention blood tests for biomarkers were conducted at a designated time in the morning (from 9 AM to 11 AM) of the second and final visits, respectively, as initial and follow-up assessments for each participant. To assess the inflammatory response and hypothalamic–pituitary–adrenal (HPA) axis activity, we measured the serum levels of interleukin 6 (IL-6), tumor necrosis factor alpha (TNF-alpha), adrenocorticotropic hormone (ACTH), and cortisol. Simultaneously, the blood level of brain-derived neurotrophic factor (BDNF) was evaluated as a neuroplasticity index. Blood samples were drawn from the antecubital vein, processed per protocol, and transported to GCCL Co., Ltd. (Gyeonggi-Do, Yongin-si, Korea), a global clinical testing laboratory. The ratios of the concentrations of IL-6 to log (BDNF) and TNF-alpha to log (BDNF) were also determined.

### 2.5. Gut Microbiome Analysis

The participants were asked to submit fecal samples at the final visit. Bacterial genomic DNA was extracted from fresh feces using a QIAamp DNA Stool Mini Kit (Qiagen, Hilden, Germany), according to a protocol by Kim et al. [31]. Genomic DNA was amplified using barcoded primers targeting the bacterial 16S rRNA V4 region gene. Each amplicon was sequenced using an Illumina iSeq 100 (Illumina, San Diego, CA, USA). Functional genes were predicted using the phylogenetic investigation of communities by the reconstruction of unobserved states (PICRUSt) [31,32]. Linear discriminant analysis (LDA) and cladograms were captured using the LDA effect size on the Galaxy platform [33]. Pyrosequencing reads were deposited in the Short Read Archive of the National Center for Biotechnology Information under accession number PRJNA 678145.

### 2.6. Safety Assessment

To evaluate the safety of probiotic administration, we monitored for the occurrence of any adverse event by participants’ self-reports at four and eight weeks after the start of the intervention. Follow-up blood cell count, biochemistry profile, urinalysis, and an electrocardiogram were also performed at the final assessment to screen for any medical problems that may have occurred during the intervention. All adverse reactions were coded to preferred terms, as specified in the Medical Dictionary for Regulatory Activities (MedDRA) version 21.0.

### 2.7. Nutritional Data

Nutrition plays a central role in the modulation of gut microbiota composition [8]. Therefore, we examined participants’ habitual dietary patterns during the intervention period. Based on 24 h dietary recalls and food diaries reported at the second, third, and final visits, the daily intakes of calories, carbohydrates, proteins, fat, water, dietary fiber, vitamins, and mineral nutrients were calculated. Nutritional data were analyzed using the Computer Aided Nutritional Analysis Program (CAN-Pro version 5.0, http://canpro5.kns.or.kr/, accessed on 24 December 2019) developed by the Korean Nu-trition Society.

### 2.8. Statistical Analysis

Sample size was determined based on the randomized controlled trial of Ali Talaei and colleagues, who adopted the BDI-II and BAI scores as primary outcome measures [32]. The minimal required sample size was calculated as 104 subjects (52 for each group) using the two-sample, continuous outcome, two-tailed, independent *t*-test approach, with a significance level of 5% and power of 80%. Considering a drop-out rate of 30%, we planned to enroll more than 150 subjects (75 per group). We performed per-protocol (PP) analysis, excluding the data of individuals who dropped out as well as those with incomplete measurements of outcomes.

Paired *t*-tests were used to evaluate intra-group differences in pre- and post-intervention assessments, including questionnaires and blood tests. Independent *t*-tests were used to evaluate inter-group differences in the changes in parameters after treatment. Effect size was calculated using Cohen’s d to estimate the magnitude of clinical improvement in each group. As for additional analysis, analysis of covariance (ANCOVA) adjusted for age, sex, baseline BDI-II score, and baseline IL-6 level was performed to compare the improvement in each subscale score of PSQI between the two groups. All statistical analyses were carried out using SAS version 9.4 (SAS Institute, Cary, NC, USA), and a two-tailed *p*-value of less than 0.05 was considered statistically significant.

## 3. Results

### 3.1. Eligibility and Baseline Characteristics

Among the 177 individuals who were invited to take part in the study, 18 were ineligible based on the inclusion or exclusion criteria, and three declined to participate. The remaining 156 individuals participated in the study and were randomly assigned (1:1 ratio) to the experimental and control groups. Seven participants withdrew consent, and 27 participants dropped out during the study period. Among the 156 participants, there were no substantial differences in the baseline demographics between those who were included or excluded from the analysis (data not shown). Ultimately, data of 122 participants (78.2% of the enrolled participants, 63 in the experimental group and 59 in the control group) were included for the per-protocol analysis (Figure 2). Table 1 indicates the baseline demographic and clinical characteristics of the two study groups and total study population. No significant differences were found in the baseline demographic and clinical characteristics between the experimental and control groups.

### 3.2. Effects of NVP-1704 on Mood

In both arms of the study, we found a substantial reduction in the SRI score at the third visit and further reduction at the final visit. Additionally, no statistically significant differences were found at the third visit (−20.27 ± 18.63 vs. −18.90 ± 21.24, *p* = 0.750) or the final visit (−27.89 ± 24.46 vs. −24.46 ± 25.95, *p* = 0.454) between the two groups (Figure 3A). We observed a significant decrease in depressive symptoms in both study groups at both four and eight weeks of intervention. Of note, those taking the NVP-1704 exhibited a greater decline in the BDI-II score compared to those on the placebo, at both the third (−6.18 ± 7.34 vs. −3.33 ± 7.03, *p* = 0.033) and final visits (−8.02 ± 7.17 vs. −5.39 ± 6.49, *p* = 0.036) (Figure 3B). The effect sizes were larger in the experimental group at both the third (Cohen’s d = 0.83 vs. 0.58) and final visits (Cohen’s d = 1.08 vs. 0.91). Regarding anxiety symptoms, we observed a more prominent decrease in the BAI score in the experimental group at the third visit (−4.73 ± 7.32 vs. −1.37 ± 7.27, *p* = 0.014) (Figure 3C). The effect size was also greater in the experimental group (Cohen’s d = 0.56 vs. 0.20). Although it did not reach statistical significance, the reduction in the BAI score tended to be more prominent in the experimental group at the final visit (−5.30 ± 8.34 vs. −2.93 ± 8.08, *p* = 0.114). Moreover, the effect size was greater in the experimental group than in the control group at the final assessment (Cohen’s d = 0.61 vs. 0.39). After additional analysis, the experimental group reported more prominent improvements in overall depression and anxiety symptoms both after four weeks (−10.90 ± 11.89 vs. −4.70 ± 12.11, *p* = 0.006) and eight weeks (−13.32 ± 13.46 vs. −8.32 ± 12.62, *p* = 0.037) of treatment (Figure 3D). Table 2 describes the changes from baseline in the SRI, BDI-II, and BAI scores in detail.

### 3.3. Effects of NVP-1704 on Sleep

Table 3 describes the changes in the PSQI and ISI scores after the intervention compared to baseline in detail. We found no significant inter-group difference between the two groups in the improvement of the PSQI score at the third visit (−0.63 ± 2.78 vs. −0.33 ± 2.54, *p* = 0.537). At the final visits, however, borderline statistical significance was observed between the two groups (−1.33 ± 3.03 vs. −0.42 ± 2.36, *p* = 0.068) (Figure 4A). In addition, the effect sizes were larger in the experimental group at both the third (Cohen’s d = 0.19 vs. 0.13) and final visits (Cohen’s d = 0.46 vs. 0.18). Those taking NVP-1704 also showed a greater magnitude of decreases in the ISI scores after eight weeks of treatment (−3.27 ± 3.89 vs. −1.14 ± 4.55, *p* = 0.006), but not after four weeks (−1.77 ± 3.95 vs. −0.43 ± 5.31, *p* = 0.123) (Figure 4B). Additionally, the effect sizes were larger in the experimental group at both the third (Cohen’s d = 0.36 vs. 0.08) and final visits (Cohen’s d = 0.75 vs. 0.25). Moreover, the control group did not show any statistically significant improvement in either the PSQI or ISI scores at the third and final visits. In contrast, the experimental group exhibited significant reductions in both the PSQI and ISI scores at the final visit.

Table 4 describes ANCOVA sub-analysis of the changes from baseline to eight weeks of treatment, in total and subscale scores of PSQI. Interestingly, those with NVP-1704 treatment represented more significance in the total PSQI scores and sleep onset latency subscale scores (*p* = 0.020 and 0.018, respectively). A greater decrease in the daytime dysfunction subscale score was observed at a borderline significance level (*p* = 0.062). Additionally, the experimental group achieved higher rates of clinically meaningful improvements in the PSQI (21/63 [33.3%] vs. 9/59 [15.3%], *p* = 0.021) and ISI scores [18/63 (28.6%) vs. 7/59 (11.9%), *p* = 0.022], compared to the control group after eight weeks of treatment (Figure 4C).

### 3.4. Blood Biomarkers

Table 5 describes changes in the level of serum markers from baseline after eight weeks of intervention for the two groups. A significant difference was identified in the changes in serum IL-6 concentrations between the two groups (−0.23 ± 1.06 pg/mL vs. 0.20 ± 1.20 pg/mL, *p* = 0.041) (Figure 5A). However, no difference was found in the change of serum BDNF levels between the two groups (Figure 5B). In contrast, the IL-6/log (BDNF) values showed a similar pattern as IL-6 levels, yielding a substantial inter-group difference (−0.02 ± 0.11 vs. 0.02 ± 0.12, *p* = 0.041) (Figure 5C). There were no significant inter- or intra-group differences following interventions in other blood biomarkers such as TNF-alpha, ACTH, cortisol, and TNF-alpha/log (BDNF) values.

### 3.5. Gut Microbiome Analysis

Figure 6 compares the gut microbiota composition between those taking NVP-1704 versus placebo. Administration of NVP-1704 tended to weakly increase α-diversity (estimated operational taxonomic unit richness, *p* = 0.223, Figure 6A), which means the abundance of strain types, and the β-diversity (principal coordinate analysis analyzed by Jensen-Shannon, Figure 6B), which means the species diversity in the microbiota composition. Figure 6C illustrates the relative contribution of major bacterial phyla in the gut microbiota composition of the two groups. NVP-1704 administration increased the *Actinobacteria* population (*p* = 0.030) and it tended to reduce the *Proteobacteria* population (*p* = 0.116), leading to a significant suppression in the ratio of *Proteobacteria* to *Actinobacteria* (*p* = 0.039) (Figure 6D). At the family level, NVP-1704 treatment significantly increased the *Bifidobacteriaceae* population (*p* = 0.047). *Lactobacillaceae* population increased with borderline significance in the experimental group (*p* = 0.078). NVP-1704 treatment also tended to reduce *Enterobacteriaceae* (belonging to *Proteobacteria*), *Muribaculaceae*, *Peptostreptococcaceae*, and *Veilonellaceae* populations (all *p* > 0.05) (Figure 6E and Appendix A). Interestingly, NVP-1704 treatment significantly reduced the ratios of *Enterobacteriaceae* to *Bifidobacteriaceae* (*p* = 0.013) and *Enterobacteriaceae* to *Lactobacillaceae* (*p* = 0.033) (Figure 6E). We found a significant increase in *Bifidobacteria*, including the *Bifidobacterium_uc* (unclassified) and *Bifidobacterium pseudolongum* strains and the *Lactobacillus reuteri* strain in the gut microbiota composition of the experimental group compared to that of the control group (*p* < 0.05 for all) (Figure 6F and Appendix A).

### 3.6. Safety and Nutritional Assessment

Safety data of 156 participants were available. No significant abnormal findings were found on the follow-up blood cell count, biochemistry profile, urinalysis, or electrocardiogram. Adverse reactions were reported by three participants (1.9%) at least once during the entire treatment period, and there were no reports of serious adverse events. The occurrence rates of adverse events were 2.6% (2/78) in the experimental arm and 1.3% (1/78) in the control arm, without significant between-group differences (*p* = 1.000). We observed only one adverse event possibly related to the intervention (gastroenteritis) in the control group, and two individuals with active treatment reported nasopharyngitis and dizziness, respectively. All adverse reactions were tolerable and self-limiting within several days, and no one discontinued treatment due to adverse events or intolerance to the products. In addition, treatment compliance assessments revealed no substantial difference between the experimental and control groups at the third [96.4% (88.6–100.0) vs. 96.8% (92.3–100.0), *p* = 0.587] or final visits [100.0% (93.8–100.0) vs. 100.0% (94.3–100.0), *p* = 0.997].

Nutritional analysis showed no significant inter-group difference in the diet composition at baseline or after treatment, except for vitamin K and magnesium (data not shown). No statistically significant difference in body weight change was found between the experimental and control groups at either the third (−0.18 ± 1.60 kg vs. 0.15 ± 1.60 kg, *p* = 0.167) or final visits (−0.03 ± 1.60 kg vs. 0.12 ± 1.12 kg, *p* = 0.363).

## 4. Discussion

The present study revealed that NVP-1704 is a safe and well-tolerated probiotic with beneficial effects on depression and sleep in healthy adults. Our study also revealed a significant reduction in serum pro-inflammatory cytokine IL-6 levels after NVP-1704 treatment. In line with these findings, our microbiome analysis demonstrated that the individuals treated with NVP-1704 had a gut microbiota composition with reduced ratios of *Enterobacteriaceae* to *Bifidobacteriaceae* and *Enterobacteriaceae* to *Lactobacillaceae*, which could be associated with better mental health. Our evaluation of safety revealed that no serious adverse reactions occurred during the study period, and the risk of adverse events with NVP-1704 treatment was low (<3%), yielding no significant difference compared to the placebo.

Our findings are similar to the results reported in a previous mouse study, which concluded that treatment with NVP-1704 led to the alleviation of depression/anxiety, changes in serum IL-6 levels, and an altered gut microbiome [17]. Therefore, the beneficial effects of NVP-1704 on depression/anxiety and the potential physiological processes underlying the psychotropic effects of NVP-1704 treatment have been implicated in both preclinical and clinical studies. We also found a significant improvement in both subjective sleep quality and insomnia symptoms after treatment with NVP-1704. To the best of our knowledge, our work is the first study to report a significant decrease in insomnia symptoms, measured using both the PSQI and ISI scales, which is attributable to probiotic treatment. A recent review of randomized controlled trials regarding the effects of probiotics on sleep reported that only a few studies revealed meaningful improvements in the PSQI score, whereas no positive changes were observed on other subjective sleep scales [34]. Although we found these clinical improvements, a non-negligible placebo effect was also observed in some of the mood and sleep parameters. The placebo effect has been frequently observed in clinical trials, especially when the evaluating parameters are subjective [35]. This study was conducted on a subclinical sample; therefore, the placebo effect could have been more pronounced.

The present study found a significant reduction in the serum pro-inflammatory cytokine IL-6 levels. The human gut microbiome is closely linked to the production of pro-inflammatory cytokines, including IL-6 [36,37]. Dysbiosis of the gut microbiome, such as a decrease in commensal *Bifidobacterium* and *Lactobacillus* strains and an increase in pathogenic gut microbes, can stimulate the secretion of pro-inflammatory cytokines via increased permeability of the intestinal epithelium [38]. In contrast, the administration of *Bifidobacterium* [39] and *Lactobacillus* strains [40] can downregulate pro-inflammatory cytokine secretion. Consistent with these results from previous studies, our findings contribute to the evidence supporting the anti-inflammatory effects of the probiotic NVP-1704.

We hypothesized that NVP-1704 treatment may increase serum BDNF levels; previous randomized clinical trials have shown that the administration of specific probiotic strains of *Bifidobacterium* and *Lactobacillus* increased serum BDNF levels [41,42,43]. In contrast to our expectation, however, we could not observe a significant difference in the change in serum BDNF levels between the two study groups. Instead, a significant decrease in the ratio of IL-6 to log (BDNF) was found after NVP-1704 treatment. A previous study suggested that an increased BDNF to IL-6 ratio can act as a surrogate maker of recovery and neuroplasticity [44]. However, because no significant difference was observed in serum BDNF levels, the changes in the ratio of IL-6 to log (BDNF) might be principally due to the impact of IL-6, not that of BDNF.

As we expected, NVP-1704 treatment induced an increase in *Bifidobacterium* and *Lactobacillus* strains and a decrease in the *Proteobacteria* population at the phylum level. Previous studies have shown that dysbiosis of the gut microbiome might be associated with psychiatric disorders, suggesting that the gut microbiome could be a useful therapeutic and preventive target. Jiang et al. [45] reported that the gut microbiota composition of patients with depressive disorders showed higher levels of *Proteobacteria* and *Enterobacteriaceae* compared to healthy controls. An altered gut microbiota profile was also reported in patients with generalized anxiety disorder [46], and gut microbiome diversity appears to be related to sleep physiology [7]. In line with previous studies, our clinical data support the notion that the oral administration of probiotic NVP-1704 can restore a balanced gut microbiota composition, thereby mitigating psychiatric symptoms.

Although the exact mechanism needs to be further elucidated, the modulation of neuroinflammatory pathways due to beneficial modification of the gut microbiome may be the key components underlying the beneficial effects of NVP-1704. Neuroinflammation induced by various medical conditions plays an important role in the pathophysiology of depression [47], anxiety [48], and insomnia [49]. Therefore, the suppression of IL-6 might be responsible for the psychotropic effects of NVP-1704 treatment.

A plethora of research has indicated that psychological stress, often represented by HPA hyperactivity, is closely related to the development of various mental health disorders, including depression [50], anxiety [51], and insomnia [52]. Hence, we hypothesized that the beneficial effects of NVP-1704 on mood and sleep may be mediated by HPA axis modulation. However, we did not detect any significant differences in either the serum ACTH or cortisol concentrations between the two groups after intervention. Although many animal studies have demonstrated that probiotics containing specific strains of *Bifidobacterium* or *Lactobacillus* modulate the HPA axis [53], clinical studies examining the effects of probiotics on HPA axis activity have revealed inconsistent findings. One study showed a correlation [14], whereas others did not [13,15]. This discrepancy may, in part, stem from methodological heterogeneity, such as the difference in the timing of blood tests (morning versus other times of the day) and differences intrinsic to the population being studied (healthy individuals versus patients with psychiatric disorders). Further methodologically robust studies are required to examine the effects of probiotics on HPA axis activity.

Our study had some limitations. First, we performed PP analysis, which may result in an overestimation of treatment efficacy and underestimation of adverse reactions, compared to intention-to-treat analysis. However, our data reported very few adverse events and good adherence (>90%), which may reduce the possibility of selection bias. Secondly, we did not analyze the gut microbiota composition prior to treatment. However, no significant differences were observed in the dietary patterns of the two study groups, suggesting that post-treatment analysis may predominantly reflect the effects of the intervention. Thirdly, we did not apply appropriate statistical corrections such as the Benjamini–Hochberg correction. We performed multiple hypothesis testing; therefore, not doing this correction may have caused false-positive findings. Additionally, we registered our study protocol to the clinical trials registry retrospectively, because Clinical Research Information Service, a Korean clinical trials registry platform, allows the retrospective registration of clinical trials. However, we should have registered the study protocol in advance, to meet international standards. In addition, a relatively large dropout rate (over 20%) may have affected the results and the interpretation of our findings. Finally, we cannot generalize our findings to the elderly aged over 65 years or to the clinical population.

## 5. Conclusions

Probiotic NVP-1704 may be helpful for alleviating subclinical symptoms of depression and anxiety in healthy adults. NVP-1704 treatment also improved sleep quality, especially sleep induction. This clinical benefit of NVP-1704 appears to stem from the restoration of a healthy gut microbiota composition, which is associated with anti-inflammatory effects. In addition, NVP-1704 treatment was well tolerated and safe, with few minor adverse events. Large-scale, highly controlled, longitudinal human studies may be conducted in the future to confirm the beneficial effects of various probiotics on mental health and sleep.

## Figures and Tables

**Figure 1 nutrients-13-02660-f001:**
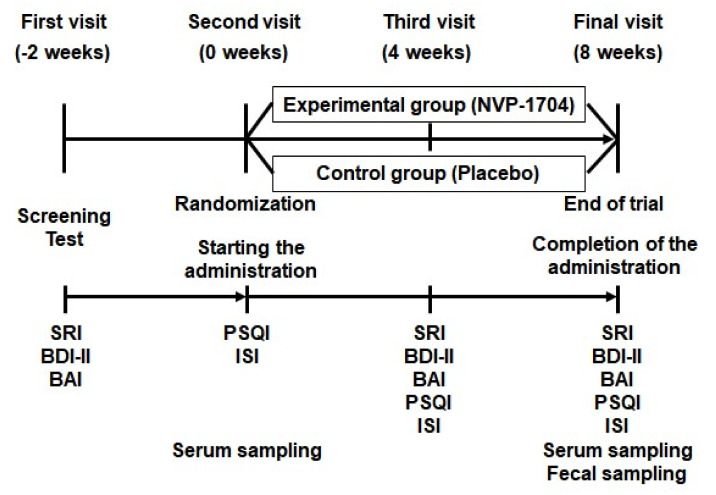
Timeline of the study protocol. SRI, Stress Response Inventory; BDI-II, Beck’s Depression Inventory-II; BAI, Beck’s Anxiety Inventory; PSQI, Pittsburgh Sleep Quality Index; ISI, Insomnia Severity Scale.

**Figure 2 nutrients-13-02660-f002:**
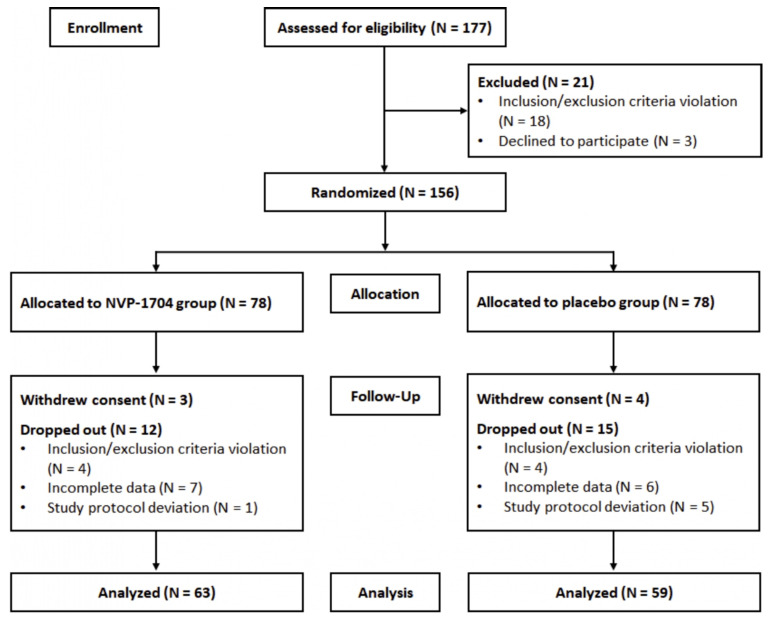
Flow chart of study participants.

**Figure 3 nutrients-13-02660-f003:**
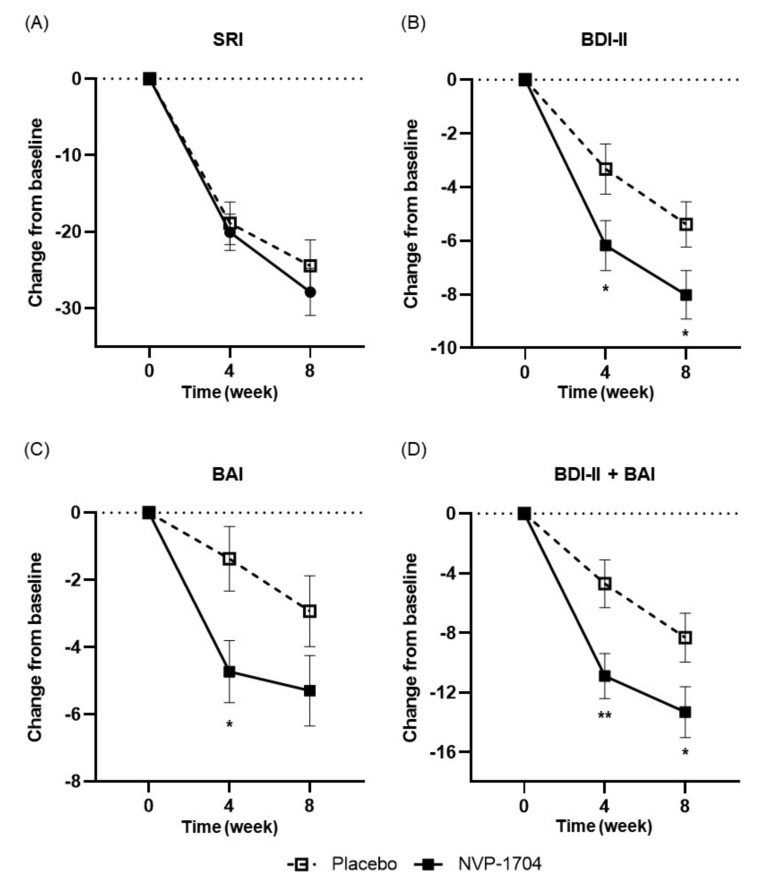
Effects of NVP-1704 on mood parameters such as SRI score (**A**), BDI-II score (**B**), BAI score (**C**), and the sum of BDI-II and BAI scores (**D**). Scores are shown as 4- and 8-week mean changes from baseline during the study period. Independent *t*-tests were performed. Data values are described as the mean ± standard error of the mean. Asterisks indicates a statistically significant difference between the two groups (* *p* < 0.05, ** *p* < 0.01). SRI, Stress Response Inventory; BDI-II, Beck’s Depression Inventory; BAI, Beck’s Anxiety Inventory.

**Figure 4 nutrients-13-02660-f004:**
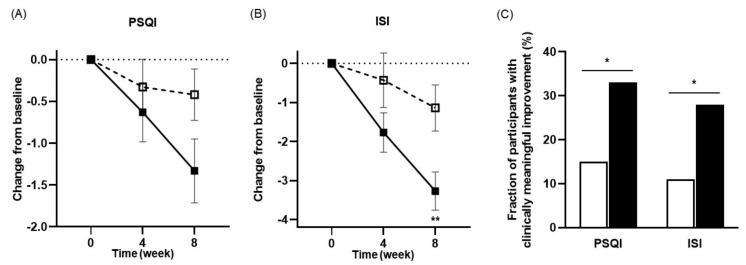
Effects of NVP-1704 on sleep parameters such as the PSQI score (**A**), ISI score (**B**), and the fraction of subjects with improvements in PSQI and ISI scores (**C**). Scores are shown as 4- and 8-week mean changes from baseline during the study period. The improvement is defined as a score reduction above or equal to the minimal clinically important difference for each scale. Independent *t*-tests were performed for analysis of the PSQI and ISI scores. Chi-squared tests were performed for the comparison of the fraction of participants with improvement. Data values are described as means ± standard errors of the mean. Asterisks indicates a statistically significant difference between the two groups (* *p* < 0.05, ** *p* < 0.01). PSQI, Pittsburgh Sleep Quality Index; ISI, Insomnia Severity Scale.

**Figure 5 nutrients-13-02660-f005:**
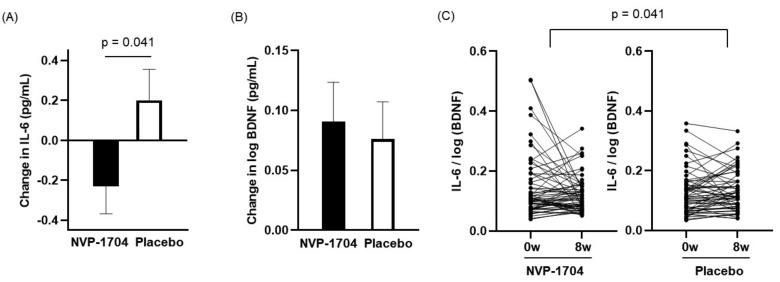
Effect of NVP-1704 on blood levels of IL-6, BDNF, and the ratio of IL-6 to log (BDNF). (**A**) Changes in the level of IL-6 and (**B**) log (BDNF) in serum from baseline to 8 weeks of treatment. (**C**) The ratio of IL-6 to log (BDNF) as individual values at each time point in the NVP-1704 and placebo groups. IL-6, interleukin-6; BDNF, brain-derived neurotrophic factor.

**Figure 6 nutrients-13-02660-f006:**
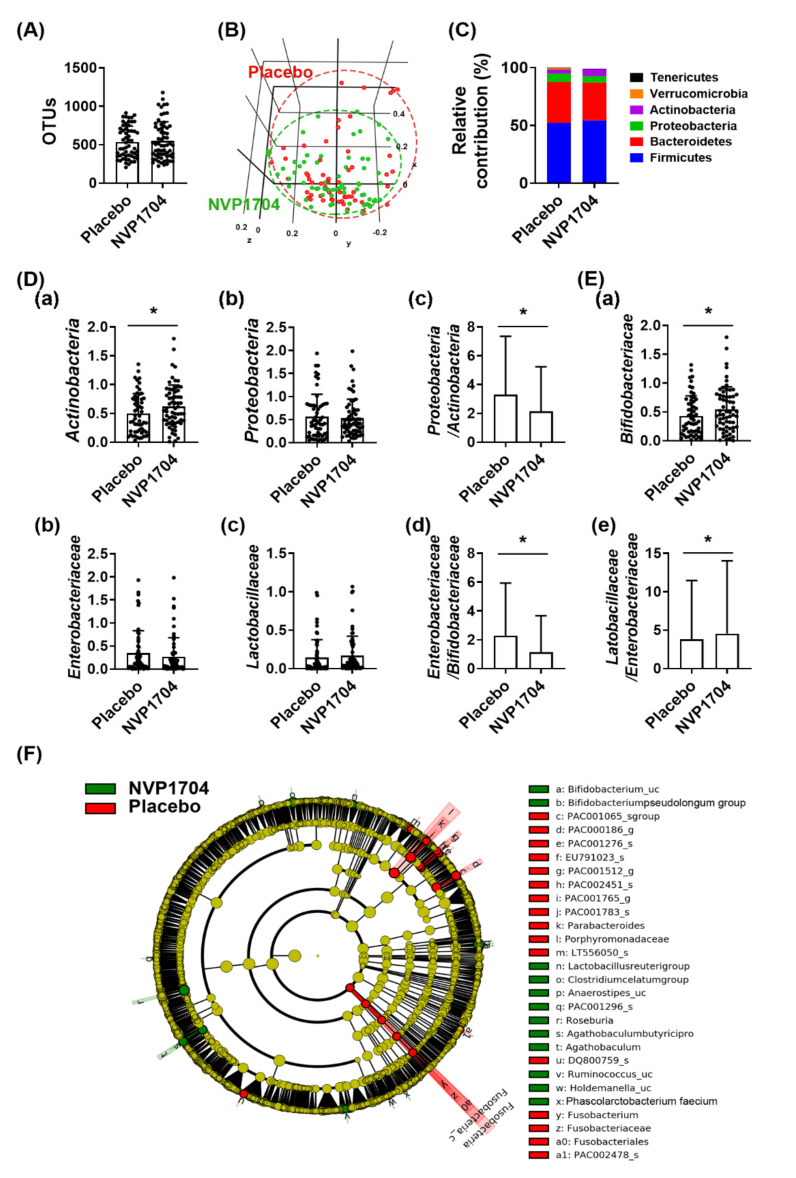
Effects of NVP-1704 on the gut microbiota composition. Effect on the α-diversity (estimated operational taxonomic units, OTUs) (**A**) and β-diversity (principal coordinate analysis [PCoA] plot based on Jansen-Shannon) (**B**). Effects on gut bacteria composition at the phylum (**C**) and cladogram (**F**) generated by linear discriminant analysis effect size indicating significant differences in gut microbial abundances. (**D**) Effect on the *Actinobacteria* (**a**) and *Proteobacteria* populations (**b**) and their ratio (**c**) at the phylum level. (**E**) Effect on the *Bifidobacteriaceae* (**a**), *Enterobacteriaceae* (**b**) and *Lactobacillaceae* populations (**c**) and as their ratio, *Enterobacteria* per *Bifidobacteriaceae* (**d**) and *Lactobacillaceae* per *Enterobacteriaceae* (**e**) at the family level. The composition of *Actinobacteria, Proteobacteria, Bifidobacteriaceae, Enterobacteriaceae,* and *Lactobacillaceae* populations was indicated as log (composition percent + 1). Data indicate mean ± standard deviation. * The significance was analyzed by one-tailed Mann–Whitney test (*p* < 0.05).

**Table 1 nutrients-13-02660-t001:** Baseline demographic and clinical characteristics of study participants.

Characteristics	Experimental Group (*n* = 63)	Control Group (*n* = 59)	Total (*n* = 122)
Sex, male	18 (28.6)	21 (35.6)	39 (31.97)
Age, years	38.86 ± 10.89	37.63 ± 11.04	38.26 ± 10.94
BMI, kg/m^2^	23.37 ± 3.31	23.44 ± 3.28	23.39 ± 3.28
Medical illness *, presence	6 (9.5)	5 (8.5)	11 (9.0)
Medication use ^†^, presence	7 (11.1)	4 (6.8)	11 (9.0)
Occupation			
Office worker	20 (31.7)	13 (22.0)	33 (27.1)
Professionals	15 (23.8)	13 (22.0)	28 (22.9)
Students	3 (4.8)	4 (6.8)	7 (5.7)
Others	6 (9.5)	8 (13.6)	14 (11.5)
No job	19 (30.2)	21 (35.6)	40 (32.8)
Physical activity ^‡^, frequency			
<1/week	28 (44.4)	25 (42.4)	53 (43.4)
1~2/week	23 (36.5)	21 (35.6)	44 (36.1)
3~4/week	9 (14.3)	11 (18.6)	20 (16.4)
>4/week	3 (4.8)	2 (3.4)	5 (4.1)
Smoking history ^§^			
Non-smoker	59 (93.7)	51 (86.4)	110 (90.2)
Ex-smoker (no smoking for > 6 months)	1 (1.6)	2 (3.4)	3 (2.5)
Smoker	3 (4.8)	6 (10.2)	9 (7.4)
Alcohol ^‖^			
No	41 (65.1)	33 (55.9)	74 (60.1)
<6 units/week	16 (25.4)	19 (32.2)	35 (28.7)
6~12 units/week	6 (9.5)	5 (8.5)	11 (9.0)
>12 units/week	0 (0.00)	2 (3.4)	2 (1.6)
Questionnaires, score			
SRI	69.57 ± 14.09	66.22 ± 12.65	67.95 ± 13.46
BDI-II	26.05 ± 7.90	23.97 ± 5.35	25.04 ± 6.83
BAI	16.35 ± 9.64	14.97 ± 7.09	15.68 ± 8.49
PSQI	8.16 ± 3.04	7.22 ± 2.29	7.72 ± 2.77
ISI	11.46 ± 4.54	10.03 ± 4.84	10.73 ± 4.76

Data are presented as the means ± standard deviations for numerical data and numbers (percentage) for categorical variables. * Medical illness includes GI disorders, metabolism and nutrition disorders, musculoskeletal and connective tissue disorders, benign and malignant neoplasm, reproductive system and breast disorders, respiratory, thoracic and mediastinal disorders, skin and subcutaneous tissue disorders, and vascular disorders in the last six months. ^†^ Medication use refers to any drug use for medical illnesses mentioned above. ^‡^ Physical activity is defined as any kind of exercise for at least 30 min. ^§^ E-cigarette use was included. ^‖^ One alcohol unit is defined as 10 mL of alcohol. BMI, body mass index; SRI, Stress Response Inventory; BDI-II, Beck’s Depression Inventory-II; BAI, Beck’s Anxiety Inventory; PSQI, Pittsburgh Sleep Quality Index; ISI, Insomnia Severity Scale.

**Table 2 nutrients-13-02660-t002:** Comparison of the changes in mood parameters from baseline to four and eight weeks of intervention between the two study groups.

Characteristics	Experimental Group (*n* = 63)			Control Group (*n* = 59)			*p* ^†^
	*n*	Mean ± SD	ES	*n*	Mean ± SD	ES	
SRI, score							
Baseline	63	69.57 ± 14.09		59	66.22 ± 12.65		
Week 4	61	49.13 ± 19.34		58	47.57 ± 18.91		
Δ		−20.27 ± 18.63	1.21		−18.90 ± 21.24	1.16	0.750
*p*-value ^‡^		<0.001			<0.001		
Week 8	63	41.68 ± 20.17		59	41.76 ± 21.27		
Δ		−27.89 ± 24.46	1.60		−24.46 ± 25.95	1.40	0.454
*p*-value ^‡^		<0.001			<0.001		
BDI-II, score							
Baseline	63	26.05 ± 7.90		59	23.97 ± 5.35		
Week 4	62	19.89 ± 6.93		57	20.47 ± 6.68		
Δ		−6.18 ± 7.34	0.83		−3.33 ± 7.03	0.58	0.033
*p*-value ^‡^		<0.001			0.007		
Week 8	63	18.03 ± 6.90		59	18.58 ± 6.46		
Δ		−8.02 ± 7.17	1.08		−5.39 ± 6.49	0.91	0.036
*p*-value ^‡^		<0.001			<0.001		
BAI, score							
Baseline	63	16.35 ± 9.64		59	14.97 ± 7.09		
Week 4	62	11.60 ± 7.16		57	13.47 ± 7.93		
Δ		−4.73 ± 7.32	0.56		−1.37 ± 7.27	0.20	0.014
*p*-value ^‡^		<0.001			0.161		
Week 8	63	11.05 ± 7.48		59	12.03 ± 7.90		
Δ		−5.30 ± 8.34	0.61		−2.93 ± 8.08	0.39	0.114
*p*-value ^‡^		<0.001			0.007		

Δ represents the change from baseline. The effect size was calculated using Cohen’s d. ^†^ Independent *t*-tests were performed to examine between-group differences in the improvement of mood symptoms. ^‡^ Paired *t*-tests were performed to examine within-group differences. Data from incomplete questionnaires were excluded from the week 4 analysis. SD, standard deviation; ES, effect size; SRI, Stress Response Inventory; BDI-II, Beck’s Depression Inventory-II; BAI, Beck’s Anxiety Inventory.

**Table 3 nutrients-13-02660-t003:** Comparison of the changes in sleep parameters from baseline to four and eight weeks of intervention between the two study groups.

Characteristics	Experimental Group (*n* = 63)			Control Group (*n* = 59)			*p* ^†^
	*n*	Mean ± SD	ES	*n*	Mean ± SD	ES	
PSQI, score							
Baseline	63	8.16 ± 3.04		59	7.22 ± 2.29		
Week 4	62	7.61 ± 2.64		58	6.90 ± 2.76		
Δ		−0.63 ± 2.78	0.19		−0.33 ± 2.54	0.13	0.537
*p*-value ^‡^		0.080			0.331		
Week 8	63	6.83 ± 2.79		59	6.80 ± 2.36		
Δ		−1.33 ± 3.03	0.46		−0.42 ± 2.36	0.18	0.068
*p*-value ^‡^		0.001			0.174		
ISI, score							
Baseline	63	11.46 ± 4.54		59	10.03 ± 4.84		
Week 4	61	9.80 ± 4.63		58	9.64 ± 4.88		
Δ		−1.77 ± 3.95	0.36		−0.43 ± 5.31	0.08	0.123
*p*-value ^‡^		0.001			0.539		
Week 8	63	8.19 ± 4.17		59	8.90 ± 4.32		
Δ		−3.27 ± 3.89	0.75		−1.14 ± 4.55	0.25	0.006
*p*-value ^‡^		<0.001			0.060		

Δ represents the change from baseline. The effect size was calculated using Cohen’s d. ^†^ Independent *t*-tests were performed to examine between-group differences in the improvement of sleep parameters. ^‡^ Paired *t*-tests were performed to examine within-group differences. Data from incomplete questionnaires were excluded from the week 4 analysis. SD, standard deviation; ES, effect size; PSQI, Pittsburgh Sleep Quality Index; ISI, Insomnia Severity Scale.

**Table 4 nutrients-13-02660-t004:** Comparison of the changes in PSQI total and subscale scores from baseline to eight weeks of intervention between the two study groups.

PSQI Subscales	Experimental Group (*n* = 63)	Control Group (*n* = 59)	*p* ^†^	*p* ^‡^
	Mean ± SD	Mean ± SD		
PSQI total				
Baseline	8.16 ± 3.04	7.22 ± 2.29		
Week 8	6.83 ± 2.79	6.80 ± 2.36		
Δ	−1.33 ± 3.03	−0.42 ± 2.36	0.068	0.020
*p*-value ^§^	0.001	0.174		
Sleep quality				
Baseline	1.60 ± 0.58	1.54 ± 0.54		
Week 8	1.38 ± 0.61	1.36 ± 0.48		
Δ	−0.22 ± 0.66	−0.19 ± 0.60	0.755	0.794
*p*-value ^§^	0.010	0.021		
Sleep latency				
Baseline	1.84 ± 0.90	1.66 ± 0.73		
Week 8	1.40 ± 0.79	1.54 ± 0.79		
Δ	−0.44 ± 0.76	−0.12 ± 0.88	0.024	0.018
*p*-value ^§^	<0.001	0.266		
Sleep duration				
Baseline	1.17 ± 1.07	0.85 ± 0.85		
Week 8	1.10 ± 0.93	0.81 ± 0.92		
Δ	−0.08 ± 1.13	−0.03 ± 0.95	0.810	0.307
*p*-value ^§^	0.578	0.784		
Sleep efficiency				
Baseline	0.43 ± 0.87	0.31 ± 0.62		
Week 8	0.32 ± 0.67	0.42 ± 0.75		
Δ	−0.11 ± 0.92	0.12 ± 0.85	0.270	0.108
*p*-value ^§^	0.340	0.290		
Sleep disturbances				
Baseline	1.35 ± 0.54	1.25 ± 0.51		
Week 8	1.21 ± 0.48	1.12 ± 0.46		
Δ	−0.14 ± 0.53	−0.14 ± 0.54	0.941	0.995
*p*-value ^§^	0.038	0.059		
Use of sleep pills				
Baseline	0.02 ± 0.13	0.02 ± 0.13		
Week 8	0.06 ± 0.30	0.08 ± 0.34		
Δ	0.05 ± 0.33	0.07 ± 0.31	0.732	0.570
*p*-value ^§^	0.260	0.103		
Daytime dysfunction				
Baseline	1.75 ± 0.76	1.59 ± 0.65		
Week 8	1.37 ± 0.79	1.46 ± 0.73		
Δ	−0.38 ± 0.99	−0.14 ± 0.75	0.125	0.062
*p*-value ^§^	0.003	0.172		

Δ represents the change from baseline. ^†^ Independent *t*-tests were performed to examine between-group differences. ^‡^ ANCOVA adjusted for age, sex, baseline BDI-II score, and baseline IL-6 level was performed to examine between-group differences in the change of the PSQI total and subscale scores. ^§^ Paired *t*-tests were performed to examine within-group differences. PSQI, Pittsburgh Sleep Quality Index; SD, standard deviation.

**Table 5 nutrients-13-02660-t005:** Comparison of the changes in blood biomarkers from baseline to eight weeks of intervention between the two study groups.

Biomarkers	Experimental Group (*n* = 63)		Control Group (*n* = 59)		*p* ^†^
	*n*	Mean ± SD	*n*	Mean ± SD	
IL-6, pg/mL					
Baseline	63	1.47 ± 1.08	59	1.29 ± 0.75	
Week 8	59	1.25 ± 0.62	59	1.49 ± 1.32	
Δ		−0.23 ± 1.06		0.20 ± 1.20	0.041
*p*-value ^‡^		0.105		0.200	
TNF-alpha, pg/mL					
Baseline	63	0.72 ± 0.32	59	0.79 ± 0.34	
Week 8	59	0.74 ± 0.20	59	0.79 ± 0.33	
Δ		0.02 ± 0.32		0.00 ± 0.32	0.704
*p*-value ^‡^		0.602		0.990	
BDNF, log pg/mL					
Baseline	63	10.02 ± 0.32	59	10.03 ± 0.31	
Week 8	59	10.10 ± 0.23	59	10.11 ± 0.27	
Δ		0.09 ± 0.25		0.08 ± 0.24	0.734
*p*-value ^‡^		0.007		0.017	
Cortisol, µg/dL					
Baseline	63	7.57 ± 4.13	59	8.28 ± 3.72	
Week 8	59	7.38 ± 2.46	59	7.69 ± 3.08	
Δ		−0.18 ± 4.50		−0.59 ± 3.89	0.597
*p*-value ^‡^		0.757		0.247	
ACTH, pg/mL					
Baseline	63	18.33 ± 14.23	59	19.29 ± 13.68	
Week 8	59	22.29 ± 13.61	59	24.23 ± 28.32	
Δ		3.66 ± 15.95		4.94 ± 23.96	0.734
*p*-value ^‡^		0.083		0.119	
IL-6/log (BDNF)					
Baseline	63	0.15 ± 0.11	59	0.13 ± 0.08	
Week 8	59	0.12 ± 0.06	59	0.15 ± 0.13	
Δ		−0.02 ± 0.11		0.02 ± 0.12	0.041
*p*-value ^‡^		0.089		0.225	
TNF-alpha/log (BDNF)					
Baseline	63	0.07 ± 0.03	59	0.08 ± 0.03	
Week 8	59	0.07 ± 0.02	59	0.08 ± 0.03	
Δ		0.00 ± 0.03		−0.00 ± 0.03	0.722
*p*-value ^‡^		0.707		0.902	

Δ represents the change from baseline. ^†^ Independent *t*-tests were performed to examine between-group differences. ^‡^ Paired *t*-tests were performed to examine within-group differences in the change of the serum levels of biomarkers. Four blood samples in the experimental group were excluded from the analysis due to errors in sample handling and transport. SD, standard deviation; IL-6, interleukin-6; TNF-alpha, tumor necrosis factor alpha; BDNF, brain-derived neurotrophic factor; ACTH, adrenocorticotropic hormone.

## Data Availability

The datasets used and/or analyzed during the current study are available from the corresponding author upon reasonable request.

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
