# Peer review of "Effects of Probiotic NVP-1704 on Mental Health and Sleep in Healthy Adults: An 8-Week Randomized, Double-Blind, Placebo-Controlled Trial"

_nutrients, 2021, doi:10.3390/nu13082660_

Round 1

Reviewer 1 Report

This is a well-designed, adequately powered, study testing the hypothesis that probiotics impact sleep and mental health in a population of healthy adults. The results were that ingestion of probiotic NVP-1704 had a greater decrease in depressive symptoms (Becks, self-report) and greater improvements sleep (Pittsburg Sleep Quality Index and Insomnia Index, self-report) than the control group. This study design was strong and outcomes clear, making this study an important contribution to the field.

Weaknesses include the following:

1) It is unclear what can be concluded from measuring serum IL-6 and BDNF. There needs to be more compelling evidence that these measures relate some real physiological process.

2) What is the justification for log transformation of the serum data and why present these values as a ratio?

3) Although there were some changes in the fecal microbial ecology based on standard 16S sequencing and taxonomy assignments, there was no effort to relate the changes to the mental outcomes, that is, sleep and affect.

4) There is a large effect of time or placebo in the control group on sleep and affect measures. This needs to be discussed.

5) the population is described as "healthy" yet the goal is to reduce depression/anxiety and sleep disturbances. The authors should better explain what healthy means.  How severe or mild is these self-reported depression/anxiety and sleep disturbances in these subjects at the study outset.  

6) What time of day were the NVP-1704 ingested?

7) It would be interesting to do quantitate the concentrations of specific ingested bacterial strains in the fecal samples using quantitative PCR.  Is there any individual differences in the amount of ingested bacteria that remains or passes rapidly through the GI.

Reviewer 2 Report

The manuscript by Lee et al. entitled "Effects of probiotic NVP-1704 on mental health and sleep in healthy adults: An 8-week randomized, double-blind, placebo-controlled trial" is an interesting piece of research. It employs randomized controlled trial (RCT) settings to examine the effect of a mixture probiotic supplementation on mental symptoms relevant to psychological stress, and some biological markers of inflammation, neurogenesis and HPA axis activity. Additionally, safety check of the intervention was performed. The paper has several strengths: it is very comprehensive, presenting multiple relevant findings; it validates the participants compliance to dietary supplementation with 16S rRNA sequencing to examine gut microbiota composition; it includes relatively large sample size. The manuscript, however, presents some problems: 1) Some results are presented in a deceptive way that exaggerate the findings (particularly that of BDNF - and inappropriate conclusion is drawn for this biomarker). 2) The study protocol  appears retrospectively registered, which should be explicitly presented and discussed as a limitation. 3) Presentation of the results is not fully clear, requires some correction. Additionally, the manuscript would benefit from adhering to the CONSORT guidelines on how to report RCT studies – eg. there is no discussion on generalizability of the findings. In the attached file I present my detailed comments (embedded into the manuscript text) to help the Authors correct their work.

Round 2

Reviewer 2 Report

The Authors improved the manuscript very much. However, the 1st round of review process revealed non-normal distribution of BDNF concentration which urges me to ask why the Authors did not perform log-transformation of this measurand values before the main BDNF analysis (as reported in Table 5). Below I present my comments in detail.

Please add the word “retrospectively” while describing the mode of trial registration in text (line 97) and in the Abstract.

Figure 1 is a great help for readers. But be coherent with abbreviations (“BDI” in figure and “BDI-II” in figure caption). Also either use “weeks” instead of “w” in the figure or explain what “w” means in the caption.

If you detected skewed distribution of BDNF concentration (as you responded to my previous comment in review 1st round regarding log-transformation), why did you not keep BDNF concentration log-transformed for the main analysis presented in Table 5? This is important to assure (relatively) normal distribution of analyzed parameters in order not to have the results biased by outlying values (if you do parametric tests).

Table 1. I did NOT mean to present data for total sample ONLY (my comment was to show ALSO the total sample data), but simply to not perform statistical tests of difference. Please keep the data for total sample and restore the separate data for experimental and control groups (see CONSORT 2010 checklist – item 15). You may also keep some information regarding similarity of baseline characteristics in the Results section, but just do not refer to “statistical testing” as it is not warranted. Additionally, in Table 1 footnote, please define what are “the unit” of alcoholic beverage, “physical activity”, “medication use” and “medical illness”,  as well as the fact of e-cigarettes inclusion – according to what you responded to my comment in review 1st round.

ANCOVA sub-analysis (line 272 and next) – please clarify that it presents the comparison in 8 week.

Figure 4 appears a bit misleading as you present the main analysis for ISI (unadjusted), but ANCOVA-adjusted analysis for PSQI total score and one of its subscale. Please include only main analyses there (although not significant for some outcomes).

Table 3 and 5. While providing the number of analyzed samples, please reveal in the footnote the reason why some samples were not analyzed (as you responded to my previous comment in review 1st round) – as it could provide some bias.

Line 305 –  “eightweeks” – please separate the words.

Figure 5A. Please take care of the lines above the figure to indicate the significance – this is not clear which bar is compared to which?

Line 336. Only borderline significance for Lactobacillaceae population – please do correct.

Line 369 – “100.0%(94.3 -103.3)” – a space is lacking. What do the values in brackets present? Higher than 100%?

Line 378-379 and 442 – referring to “IL-6 to log (BDNF) ratio” is not needed as it is further extension of IL-6 results and may suggest to readers the BDNF changes between the groups.

Lines 382-382. Keep italics for family names of bacteria.

Lines 417-425. While discussing IL-6/log(BDNF) ratio, please specify that the impact of IL-6 on the ratio appears much higher than that of BDNF. In this form, the discussion may be regarded as appreciation of some impact of BDNF, which was indeed insignificant between groups (p=0.859).

Line 480. Please avoid referring to “neuroprotective” effect due to no significant result for BDNF.
